

# Online measurement of highly oxygenated compounds from organic aerosol

Ella Häkkinen[1], Jian Zhao[1], Frans Graeffe[1], Nicolas Fauré[2], Jordan E. Krechmer[3], Douglas Worsnop[3], Hilkka Timonen[4], Mikael Ehn[1], and Juha Kangasluoma[1]

[1]Institute for Atmospheric and Earth System Research/Physics, Faculty of Science, University of Helsinki, Helsinki, 00140, Finland.
[2]Department of Chemistry and Molecular Biology, Atmospheric Science, University of Gothenburg, SE-41296 Gothenburg, Sweden.
[3]Aerodyne Research Inc., Billerica, Massachusetts, 01821, United States.
[4]Atmospheric Composition Research, Finnish Meteorological Institute, 00560 Helsinki, Finland.

**Correspondence:** Ella Häkkinen (ella.hakkinen@helsinki.fi)

**Abstract.** Highly oxygenated compounds are important contributors to the formation and growth of atmospheric organic aerosol, and thus have an impact on Earth's radiation balance and global climate. However, knowledge of the contribution of highly oxygenated compounds to organic aerosol and their fate after condensing into the particle phase has been limited by the lack of suitable detection techniques. Here, we present a new online method for measuring highly oxygenated compounds from

organic aerosol. The method includes thermal evaporation of particles in a new inlet, Vocus inlet for aerosols (VIA), followed by identification of the evaporated highly oxygenated compounds by a nitrate chemical ionization mass spectrometer ($NO_3$-CIMS). The method does not require sample collection, enabling highly time-resolved measurements of particulate compounds. We evaluate the performance of the method by measuring the detection limit and performing background measurements. We estimate a detection limit of below 1 ng m$^{-3}$ for a single compound and below 1 μg m$^{-3}$ for SOA with the sampling set-up

used here. These detection limits can be improved upon by optimizing the flow setup. Furthermore, we detect hundreds of particulate highly oxygenated compounds from organic aerosol generated from different precursors. Our results are consistent with previous studies showing that the volatility of organic compounds decreases with increasing *m/z* ratio and higher level of oxygenation, and that organic aerosol consists of monomers and oligomeric compounds. By comparing the gas- and particle-phase compounds, we found indications of potential particle-phase reactions occurring in organic aerosol. Future work will

focus both on further improving the sampling design, as well as on better understanding the evaporation dynamics of the system, but already these initial tests show that VIA coupled to the $NO_3$-CIMS is a promising method for investigating the transformations and fate of the compounds after condensing into the particle phase.

## 1  Introduction

Atmospheric aerosol particles influence Earth's climate by scattering or absorbing solar radiation and by altering cloud prop-

erties (Carslaw et al., 2013). Secondary organic aerosol (SOA), formed in the atmosphere from gas-to-particle conversions, contributes a large fraction to the submicron tropospheric aerosol (Jimenez et al., 2009). SOA is produced by the oxidation of





volatile organic compounds (VOCs) and it is a complex system consisting of many individual organic compounds partitioning between the gas and the particle phase. Only a small fraction of the compounds within SOA are identified, leading to large uncertainties in estimating the climate impacts of SOA (Hallquist et al., 2009; Glasius and Goldstein, 2016).

Recently, it was discovered that some VOCs form highly oxygenated organic molecules (HOM) via a process called autoxidation (Crounse et al., 2013). These molecules are estimated to explain a large fraction of SOA formation due to their low volatilities (Ehn et al., 2014; Kirkby et al., 2016). While HOM are extensively studied and observed in the gas phase, their fate after condensing into the particle phase is poorly characterized due to limitations in the measurement techniques. Simply studying the condensing compounds is not enough, owing to potential chemical reactions taking place within the particles. These

reactions can lead to changes in the chemical composition and properties of SOA, thus affecting its formation and evolution in the atmosphere. It has been suggested that after condensation, HOM either remain in the particle phase without structural changes, undergo fragmentation reactions or form larger components of SOA (Mutzel et al., 2015; Bianchi et al., 2019). To improve our understanding of the aerosol formation and growth processes that influence Earth's radiation balance and global climate, a suitable method for measuring the highly oxygenated compounds from organic aerosol is needed.

Chemical ionization mass spectrometry is the main tool for the detection of gas-phase oxygenated organic compounds (Lee et al., 2014; Ehn et al., 2014; Riva et al., 2019b). In order to make the particle-phase molecules detectable with a mass spectrometer, they need to be converted to gas-phase ions. An offline or semi-online thermal desorption methods are commonly used to measure particle-phase compounds. Smith et al. (2004) presented the thermal desorption chemical ionization mass spectrometer (TDCIMS) for measuring the composition of sub-20 nm aerosol. They use an aerosol charger to collect ambient

particles on a wire in an electrostatic precipitator, which is then heated and the evaporated compounds are analyzed with a chemical ionization mass spectrometer. Another instrument utilizing particle collection, the filter inlet for gases and aerosol (FIGAERO) (Lopez-Hilfiker et al., 2014), is used to analyze the aerosol composition by collecting sample aerosol on a PTFE filter and desorbing the collected sample with heated $N_2$ flow. In addition to these, thermal desorption is also utilized in the chemical analysis of aerosol online (CHARON) inlet (Eichler et al., 2015). It consists of a gas-phase denuder for removal of the

gas-phase compounds, an aerodynamic lens for particle collimation and a thermodesorption unit for particle evaporation. The evaporated compounds can be analyzed with low-pressure gas analyzers. Recently developed extractive electrospray ionization time-of-flight mass spectrometer (EESI-TOF) (Lopez-Hilfiker et al., 2019), is an online method for aerosol analysis without thermal desorption. In EESI-TOF, the sample particles collide with charged electrospray droplets and the soluble compounds are extracted and ionized as the electrospray solvent evaporates.

Here, we present an online thermal evaporation method for measuring highly oxygenated compounds from organic aerosol. The method includes a new inlet, Vocus inlet for aerosols (VIA), for particle evaporation and a nitrate chemical ionization mass spectrometer ($NO_3$-CIMS) to detect the evaporated highly oxygenated compounds. The method does not require sample collection, enabling highly time-resolved measurements of particulate compounds. Furthermore, the VIA can be coupled to different gas-phase analyzers and aerosol instruments. In this study we first evaluate the performance of the method and then

we demonstrate the detection of particulate highly oxygenated compounds by applying the method to laboratory-generated organic aerosol.



## 2 Methods

### 2.1 Experimental set-up

We performed a series of laboratory experiments to test the performance of the new method with an experimental set-up shown
in Fig. 1. In order to produce aerosol, either an environmental chamber, an oxidation flow reactor, or an atomizer was used.
The chamber experiments were performed using $\alpha$-pinene as a SOA precursor, whereas the flow reactor experiments were
performed with different SOA precursors by injecting a specific VOC into the reactor. We used two monoterpenes ($C_{10}H_{16}$)
$\alpha$-pinene and $\beta$-pinene, a sesquiterpene $\beta$-caryophyllene ($C_{15}H_{24}$), and an alkane n-decane ($C_{10}H_{22}$) as SOA precursors.
The atomizer was used to produce sucrose particles from sucrose water solution. We used sucrose ($C_{12}H_{22}O_{11}$) to produce
more chemically simplified aerosol than SOA. The generated aerosol particles were directed through VIA, a new inlet that
is discussed in detail in Sect. 2.1.3, in which the gas-phase compounds were removed from the sample air and the thermal
evaporation took place. The evaporated highly oxygenated compounds were detected by a $NO_3$-CIMS and the particle size
distribution was measured by a scanning mobility particle sizer (SMPS). A high efficiency particulate air (HEPA) filter in front
of the VIA was used to remove particles from the sample air for background measurements. The bypass line was used to bypass
the VIA and measure the non-heated sample air directly by the instruments.

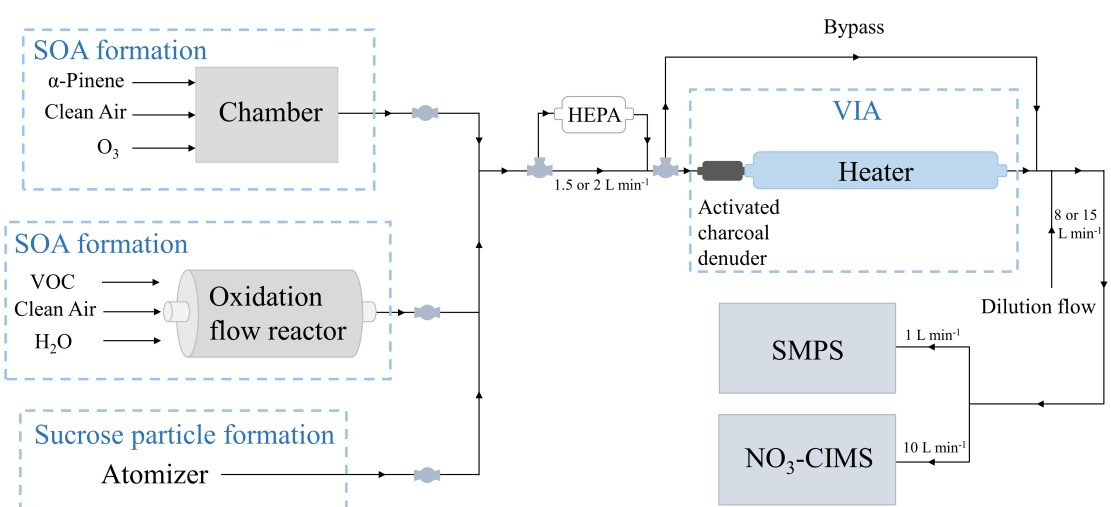

**Figure 1.** Schematic of the experimental set-up.



### 2.1.1 Oxidation flow reactor

We used a potential aerosol mass (PAM) oxidation flow reactor (OFR) (Kang et al., 2007; Lambe et al., 2011) in order to oxidize VOCs into low volatility vapors that form SOA. The aim was to generate aerosol for characterization of the VIA-NO$_3$-CIMS system, not to investigate the aerosol aging inside the flow reactor. The oxidation flow reactor is made of stainless steel and has a volume of approximately 13 liters. The UV lamps inside the reactor operate at wavelengths 185 and 254 nm, producing O$_3$ from O$_2$ and OH and HO$_2$ from H$_2$O. This highly oxidizing environment allows simulations of atmospheric oxidation processes at accelerated rates (Kang et al., 2007). We used dry air purified with a clean air generator (AADCO, series 737-14, Ohio, USA) as a carrier gas and to produce O$_3$ in the flow reactor. In some of the experiments we injected water vapor (RH 22 to 33 %) into the flow reactor to produce OH. The total flow through the flow reactor was 10 L min$^{-1}$, resulting in a ∼ 1 min residence time. The voltage of the UV lamps, controlling the intensity of the lamps, was varied between 100 V and 130 V. We generated different types of SOA by injecting four different VOCs ($\alpha$-pinene, $\beta$-caryophyllene, $\beta$-pinene and n-decane) separately into the flow reactor. The VOCs were injected continuously into the flow reactor using a syringe pump at a constant rate.

### 2.1.2 Environmental chamber

In addition to the oxidation flow reactor, we used an environmental chamber to generate SOA to define the detection limit of the VIA-NO$_3$-CIMS system and to investigate the differences between the gas- and the particle-phase compounds. The chamber is made of Teflon and it is 2 m$^3$ in volume. Detailed information about the chamber is given in Riva et al. (2019a) and Peräkylä et al. (2020). We injected clean air, ozone generated by an ozone generator (Dasibi 1008-PC), and $\alpha$-pinene continuously into the chamber with a total flow of 40 L min$^{-1}$. The residence time in the chamber was ∼ 50 min. To increase the SOA mass loading in the chamber, we used inorganic seed particles to provide surface for the gas-phase compounds to condense on. We used sodium chloride seed particles instead of commonly used ammonium sulfate to minimize evaporation from the seed particles in the VIA. The seed particles were generated by atomizing sodium chloride water solution, and selecting 80 nm particles with a differential mobility analyzer. We measured the $\alpha$-pinene ozonolysis gas-phase compounds by sampling directly from the chamber without dilution, and the particle-phase compounds with VIA according to Fig. 1.

### 2.1.3 VIA

The VIA consists of an activated charcoal gas denuder and a thermal desorption unit. The denuder (Eichler et al., 2015) removes the gas-phase compounds from the sample air, and the thermal desorption unit initiates aerosol evaporation. The desorption unit is 40 cm in length and it is made of 1/4" Sulfinert coated stainless steel tubing. The VIA is an inlet originally designed for Vocus proton-transfer-reaction time-of-flight mass spectrometer (PTR-TOF) for online analysis of aerosol composition (Aerodyne Research Inc.). However, in this work we adapted the VIA to interface with a NO$_3$-CIMS. In our experiments, the temperature of the thermal desorption unit was set between 25 - 300 °C. To maintain the performance of the activated charcoal denuder, we regenerated it with heated clean air after each laboratory experiment. The total flow through the VIA was 1.5 L



$\mathrm{min}^{-1}$ in the oxidation flow reactor experiments and 2 L $\mathrm{min}^{-1}$ in the chamber experiments, resulting in residence times of 0.5 s and 0.38 s inside the thermal desorption unit of the VIA when not accounting thermal expansion effects. After the VIA, a dilution flow of 8 or 15 L $\mathrm{min}^{-1}$ of clean air was added to cool down the sample flow before entering the instruments and to provide sufficient flow for the instruments.

## 2.2 Instrumentation

### 2.2.1 NO$_3$-CIMS

A nitrate ion based chemical ionization mass spectrometer (NO$_3$-CIMS) is highly selective towards oxygenated compounds, and thus we used it to detect the evaporated highly oxygenated compounds from the particle phase. The instrument consists of an inlet for chemical ionization (CI) and an atmospheric pressure interface time-of-flight (APi-TOF) high resolution mass spectrometer (Junninen et al., 2010; Jokinen et al., 2012). The sheath flow of the chemical ionization inlet containing nitric acid (HNO$_3$) is guided through an x-ray source, where HNO$_3$ molecules are ionized to nitrate ions. The nitrate ions are directed into the sample flow where they ionize the sample molecules either via proton transfer reaction or via clustering the sample molecule to the nitrate ion (Jokinen et al., 2012; Hyttinen et al., 2015). Highly oxygenated compounds are primarily detected as clusters with NO$_3^-$ according to Eq. (1).

$$NO_3^- + X \rightleftharpoons NO_3^- \cdot X \tag{1}$$

After ionization, sample ions are guided into the APi-TOF, where their signals and mass-to-charge (*m/z*) ratios are determined. The signal intensity of highly oxygenated compounds can be converted to concentration according to Eq. (2) (Jokinen et al., 2012).

$$[X] = C_x \cdot \frac{(X \cdot NO_3^-)}{\sum_{i=0}^{2}(HNO_3)_i(NO_3^-)} \tag{2}$$

where [X] is the concentration of the measured compound, the numerator describes the signal of compound X clustered with nitrate ion and the denominator describes the sum of reagent ion signals. $C_x$ is a calibration factor, which is determined in Sect. 3.2.

The instrument was sampling after the VIA with a flow rate of 10 L $\mathrm{min}^{-1}$, through a $\sim$ 1 m long tubing. Data were recorded with a time resolution of 10 s. We processed the NO$_3$-CIMS data using tofTools software package (Junninen et al., 2010).

### 2.2.2 SMPS

We used a scanning mobility particle sizer (SMPS) to measure the particle size distribution before and after the particles were heated in the VIA. SMPS consists of a radioactive charger (Ni-63) to charge the particles, a differential mobility analyzer (DMA) to classify the particles according to their electrical mobility diameter and a condensation particle counter (CPC;



Model 3772 and 3750, TSI) to count the particle number concentration. The SMPS measured particles from 10 nm to 500 nm in diameter with a scanning time of $\sim 130$ s and a sample flow rate of 1 L min$^{-1}$.

## 3    Performance and characterization of the method

### 3.1    Temperature profile and transmission efficiency of the VIA

The VIA temperature is set with respect to the measured temperature values from the outer surface of the thermal desorption unit. Thus, the accurate internal temperature value is not known. To investigate the temperature profile inside the thermal desorption unit, we placed a thermocouple into different locations inside the unit and measured the temperature of the gas at different set temperatures and flow rates. The temperature profile inside the unit was not uniform, as the gas was colder at the entrance than at the outlet of the unit, where the temperature reached the setpoint. The temperatures reported in this study are

the setpoint values of the thermal desorption unit of the VIA, not necessarily the actual internal temperatures.

In our experiments, the residence time inside the desorption unit was 0.38 s or 0.5 s, and the time that the particles were exposed to the set temperature at the outlet of the unit was even shorter than the residence time. During the calibration of the NO$_3$-CIMS (discussed in the following section), we measured that around 70 % of the ammonium sulfate mass was evaporated at set VIA temperature of 300 °C. However, previous thermal denuder studies have shown that ammonium sulfate particles

completely evaporate at 180 °C (Huffman et al., 2008; Hakala et al., 2017). This suggests that the time particles spent at the set temperature was too short for complete particle evaporation and a fraction of the mass remained on the particles.

The particle transmission efficiency of the VIA was measured by sampling indoor air through the VIA and measuring the transmitted particle concentration with a condensation particle counter (CPC). The VIA was set to 25 °C and the total flow through the VIA was 1 L min$^{-1}$. About 90 % of indoor particles with diameter ranging from 10 nm to 200 nm (mode at 20

nm) transmitted through the VIA.

### 3.2    Calibration of the NO$_3$-CIMS and detection limit of the method

A simple method to calibrate the NO$_3$-CIMS for highly oxygenated compound detection does not exist due to a lack of suitable standards. We used sulfuric acid (H$_2$SO$_4$) to determine the calibration factor, assuming that the sensitivities of nitrate clusters to sulfuric acid and highly oxygenated compounds are equal. We generated ammonium sulfate ((NH$_4$)$_2$SO$_4$) particles by

atomizing ammonium sulfate water solution. The particles were heated in the VIA set to 300 °C and the SMPS measured the size distribution of ammonium sulfate particles before and after the particles were heated. The mass concentration of evaporated sulfuric acid was calculated from the size distribution data, assuming spherical particles of density of 1770 kg m$^{-3}$. The fraction of sulfuric acid in ammonium sulfate particles was considered in the calculation. The NO$_3$-CIMS detected the evaporated sulfuric acid (H$_2$SO$_4$) as H$_2$SO$_4$NO$_3^-$ and HSO$_4^-$ ions, and the sum of their signal intensities was converted into

mass concentration by applying different calibration factors (Fig. 2). The decreasing trend seen in the calibration curves could be due to nucleation under high concentration of sulfuric acid. On average, the difference between the concentrations measured



by the SMPS and the NO$_3$-CIMS is smallest when calibration factor of $2\times10^{10}$ cm$^{-3}$ is used. Therefore, we decided to use the calibration factor of $2\times10^{10}$ cm$^{-3}$ when converting the NO$_3$-CIMS signals to concentrations. We stress here that this value provides concentrations of highly oxygenated compounds with large uncertainties. Hence, we focus more on the qualitative

than on the quantitative analysis of the measured data.

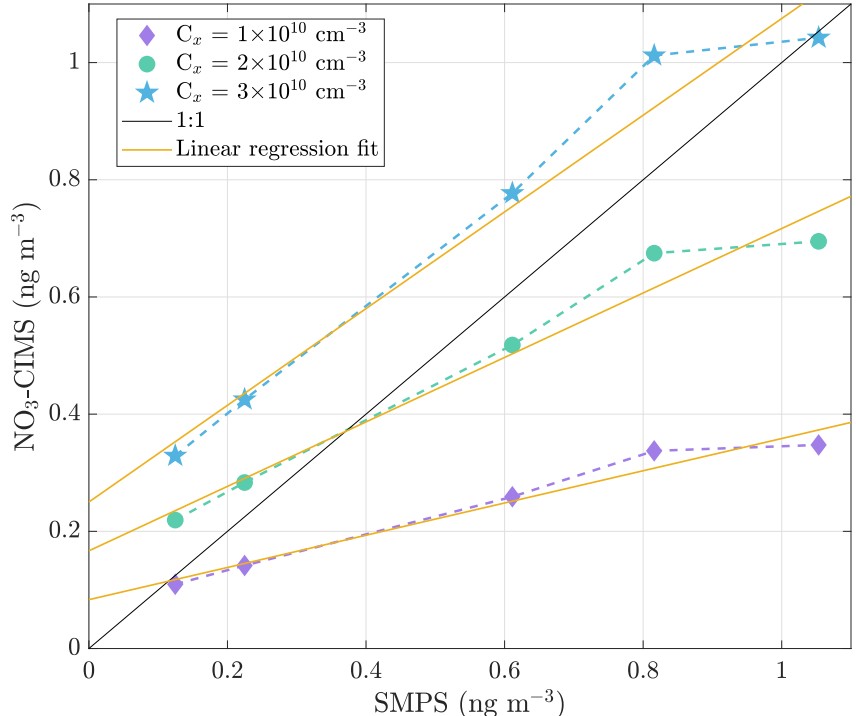

**Figure 2.** Correlation between the mass concentration of evaporated sulfuric acid measured with the SMPS and the NO$_3$-CIMS. The signal intensities measured by the NO$_3$-CIMS were converted into mass concentration by applying calibration factors of $1\times10^{10}$, $2\times10^{10}$ and $3\times10^{10}$ cm$^{-3}$. 1:1 line represents the points where the corresponding concentrations are identical. The solid yellow lines show the linear regression fits to the calibration curves.

We fit a linear regression model to the calibration curves, resulting in y-intercepts of 0.08, 0.17 and 0.25 ng m$^{-3}$ for the calibration factors of $1\times10^{10}$, $2\times10^{10}$ and $3\times10^{10}$ cm$^{-3}$, respectively (Fig. 2). Based on the calibration data, we estimate detection limit of below 1 ng m$^{-3}$ for a single compound (H$_2$SO$_4$). To evaluate the detection limit for bulk organic aerosol, we performed experiments in the environmental chamber. We added ozone (33 ppb) and $\alpha$-pinene (120 ppb) to the chamber

and observed aerosol formation and growth by the SMPS as shown in Fig. 3. During this event, the VIA sampled from the chamber at set temperature of 230 °C and the NO$_3$-CIMS measured after the VIA. The sample flow through the VIA was 2 L min$^{-1}$ and a dilution flow of 8 L min$^{-1}$ of clean air was added after the VIA. When the SOA mass loading in the chamber was





below 0.5 µg m$^{-3}$, the NO$_3$-CIMS signals for compounds 200-750 *m/z* started to increase. Therefore, with the crude sampling set-up used here, we found a detection limit of well below 1 µg m$^{-3}$ for organic aerosol. This is substantially higher compared
to the detection limit for sulfuric acid. Organic aerosol potentially contains thousands of different compounds and its signal is spread out over wider *m/z* range than the signal of sulfuric acid, mainly explaining the difference between the detection limits. Furthermore, the NO$_3$-CIMS can not detect all of the compounds evaporating from SOA, whereas it can detect majority of the evaporated sulfuric acid. The VIA-NO$_3$-CIMS system can be operated in different ways and the detection limits obtained in this study are for the particular experimental set-up presented in Sect. 2.1. The detection limit could be improved upon by
optimizing the sampling set-up, such as increasing the sample flow by using several denuders in parallel and optimizing the mixing of the sample and the dilution flow.

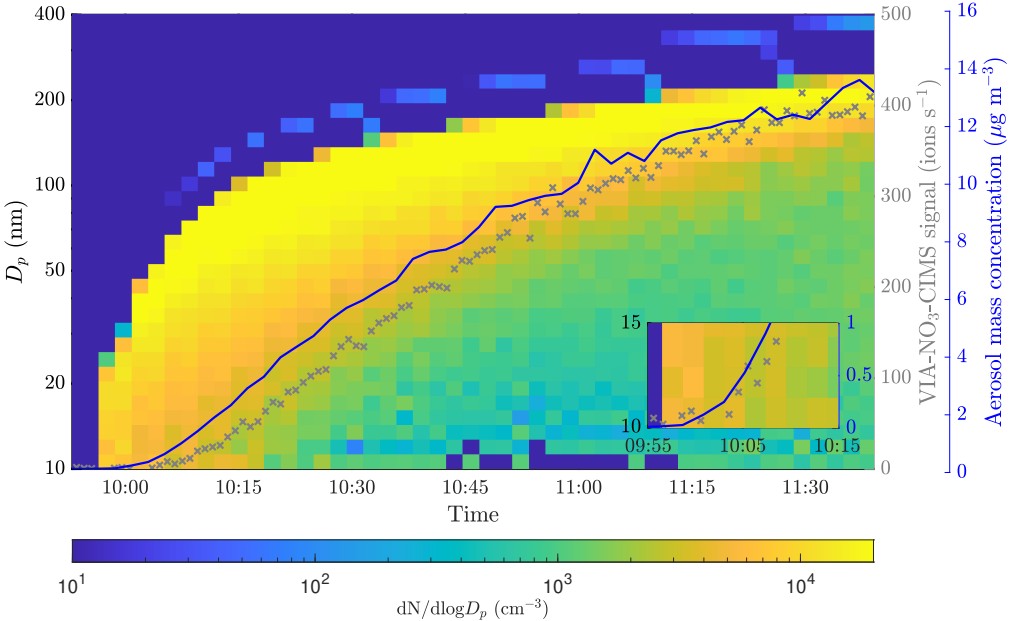

**Figure 3.** SOA formation and growth from $\alpha$-pinene ozonolysis observed by the SMPS. The blue line shows the aerosol mass concentration measured by the SMPS. The grey markers show the sum of the NO$_3$-CIMS signals for compounds from 200 to 750 *m/z* sampled through the VIA set to 230 °C. The inset shows a zoomed region of the point where the NO$_3$-CIMS started to detect compounds.

### 3.3 Background measurements

To show that the compounds detected by the NO$_3$-CIMS are evaporated from the particle phase, several background measurements were conducted. Figure 4 shows a comparison between the background spectra and the particle-phase spectra measured by the NO$_3$-CIMS. The background spectra (4a,b,c,e) show completely different pattern than the particle-phase spectra (4d and





f), indicating that the compounds detected by the NO$_3$-CIMS are evaporated from the particle phase. The background spectrum measured through HEPA filter (4e) was recorded for each set VIA temperature (25 °C, 70 °C, 120 °C, 170 °C, 230 °C and 300 °C) and subtracted from the SOA evaporation spectra at the sampled temperature.

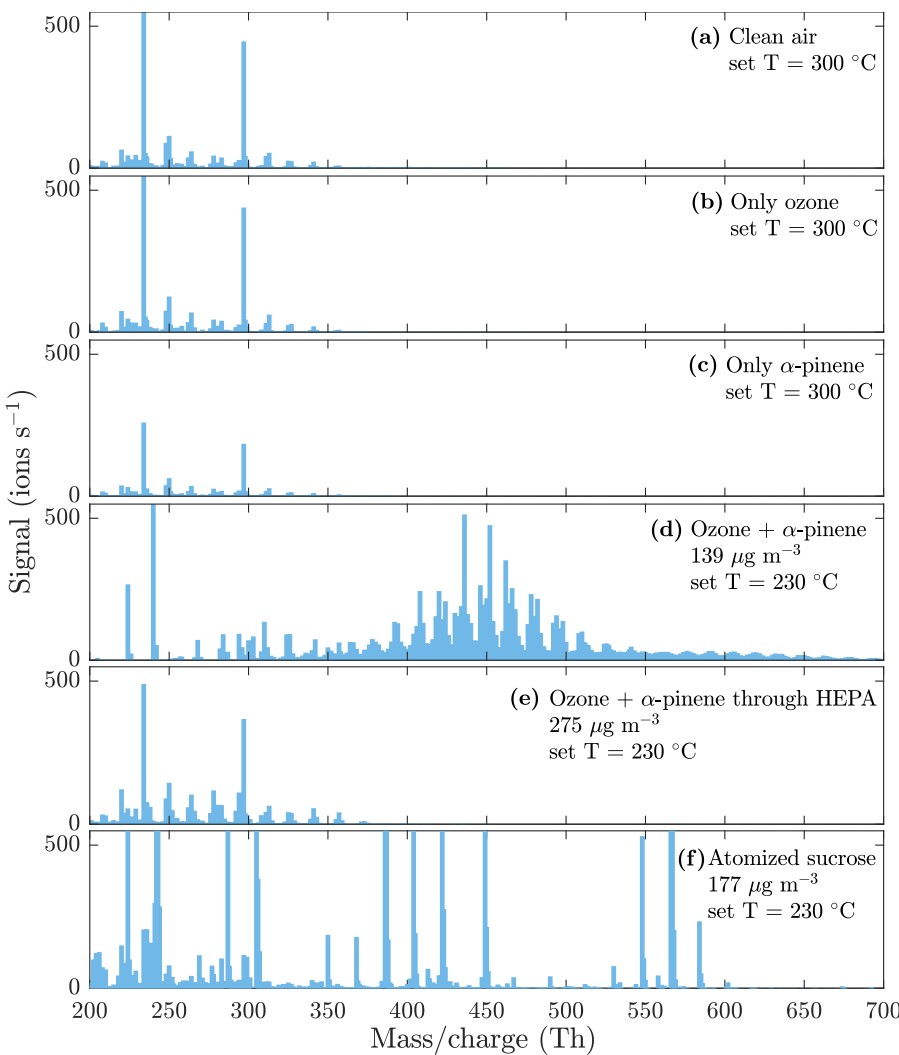

**Figure 4.** Evaporation spectra measure by the NO$_3$-CIMS during the heating of (a) only clean air (b) only ozone (c) only $\alpha$-pinene (d) $\alpha$-pinene SOA (e) $\alpha$-pinene SOA with a HEPA filter added in front of the VIA (see Fig. 1 for the placement of the HEPA filter) and (f) sucrose particles. The sample flow through the VIA was 1.5 L min$^{-1}$ and a dilution flow of 15 L min$^{-1}$ of clean air was added after the VIA. The concentrations are the total aerosol mass loadings in the VIA before dilution. A peak list of the main compounds evaporated from $\alpha$-pinene SOA in panel (d) is provided in the Appendix (Table A1).



## 4  Results and discussion

### 4.1  Application to organic aerosol

To demonstrate the detection of particulate highly oxygenated compounds with the VIA coupled to the NO$_3$-CIMS, we applied the described method to SOA generated from different precursors. The selected precursor reacted with ozone in the oxidation flow reactor, leading to SOA mass loading of 60 to 500 µg m$^{-3}$ that was sampled through the VIA. The VIA temperature was set to six different temperature stages between 25 and 300 °C and data were recorded for 10 minutes for each temperature. The sample flow through the VIA was 1.5 L min$^{-1}$ and a dilution flow of 15 L min$^{-1}$ of clean air was added after the VIA.

During the heating of SOA formed from $\alpha$-pinene ozonolysis, the NO$_3$-CIMS detected oxygenated organic compounds evaporating from the particle phase (Fig. 5). These compounds can be divided into three classes according to their *m/z* ratio: monomers (240-370 Th), dimers (410-550 Th), and trimers (560-700 Th). Monomers include mainly $C_{9-10}$ compounds, dimers include mainly $C_{19-20}$ compounds and trimers include mainly $C_{29-30}$ compounds clustered with a nitrate ion. To better show the trimer region, we include the $\alpha$-pinene ozonolysis SOA evaporation spectra with a logarithmic y-scale in the Appendix (Fig. A1). The oxygen content of the identified evaporated compounds varies between 4 and 15 oxygen atoms. Our results show that monomers start to evaporate at set temperature of 70 °C, dimers at set temperature of 120 °C, and trimers at set temperature of 230 °C. The removal of the gas-phase compounds before the VIA disturbs the gas-particle equilibrium, potentially leading to evaporation and detection of some organic compounds (mainly C$_5$ compounds) already at 25 °C. According to the SMPS data, roughly half of the SOA mass have evaporated at 120 °C and almost all (> 99%) of the mass have evaporated at 300 °C. The dominating signals are C$_{10}$H$_{16}$O$_7$, C$_{10}$H$_{16}$O$_6$, and C$_{10}$H$_{14}$O$_8$ for monomers, C$_{18}$H$_{30}$O$_8$, C$_{18}$H$_{30}$O$_9$, and C$_{20}$H$_{32}$O$_8$ for dimers and C$_{30}$H$_{42}$O$_8$ and C$_{30}$H$_{40}$O$_9$ for trimers. A peak list summarizing the elemental compositions and the *m/z* ratios of the compounds evaporated with the highest signal intensities is provided in the Appendix (Table A1). Our findings are consistent with previous studies showing that the volatility of organic compounds decreases with increasing *m/z* ratio and higher level of oxygenation (Kroll and Seinfeld, 2008; Peräkylä et al., 2020) and SOA consisting of oligomeric compounds (Lopez-Hilfiker et al., 2015; Zhang et al., 2017; Pospisilova et al., 2020). The compounds detected with the NO$_3$-CIMS were identified using high resolution analysis in order to separate the individual ions present at the same unit mass, as shown in Fig. 6. The signal intensities of trimers and their signal-to-noise ratios are lower compared to monomers and dimers, yet trimer signals are clearly distinguishable from the noise level. The majority of the monomers and dimers identified here have been detected in previous studies either in the gas phase or in the particle phase (Ehn et al., 2014; Quéléver et al., 2019; Pospisilova et al., 2020), whereas the trimers have not previously been observed in either phases. Our findings suggest that the particulate compounds that have not previously been found from the gas phase are formed through particle-phase reactions or through some other processes, such as thermally induced processes within VIA.





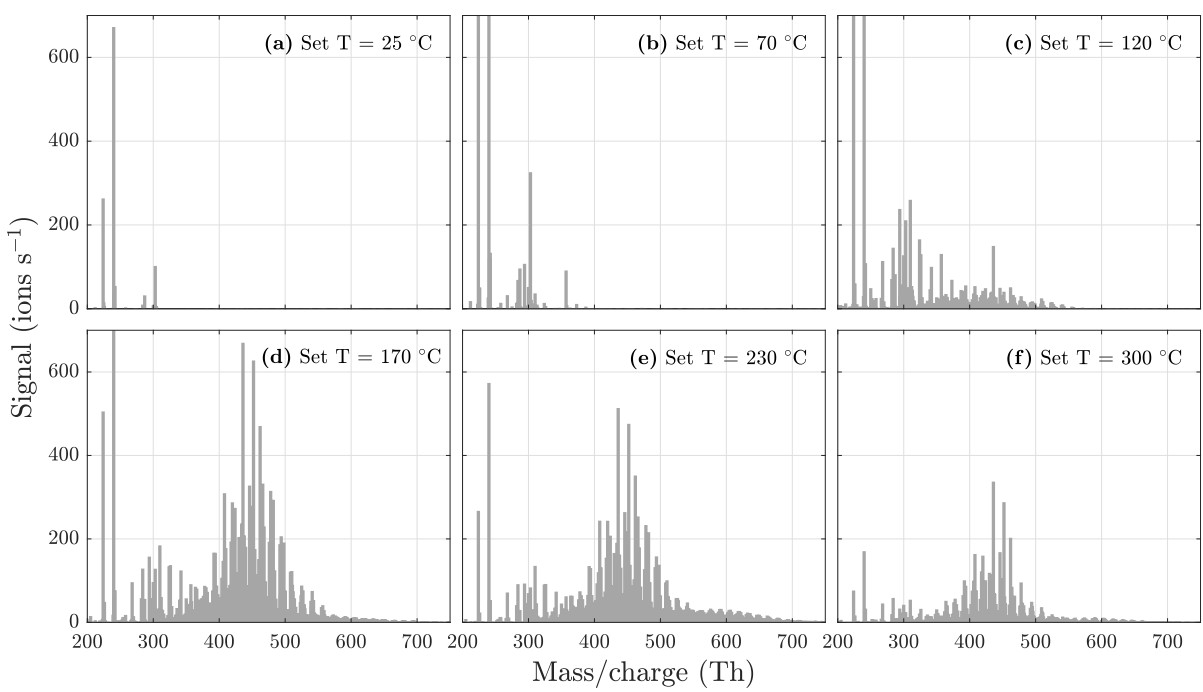

**Figure 5.** NO$_3$-CIMS spectra at 1 min averaging measured during $\alpha$-pinene ozonolysis SOA evaporation at different temperatures. The total SOA mass loading was 139 µg m$^{-3}$ in the VIA before dilution. Note that the y-axes are set to the same scale, and therefore the full peaks at *m/z* 224 and 240 are not visible in each spectrum. All molecules were detected as clusters with the nitrate ion.





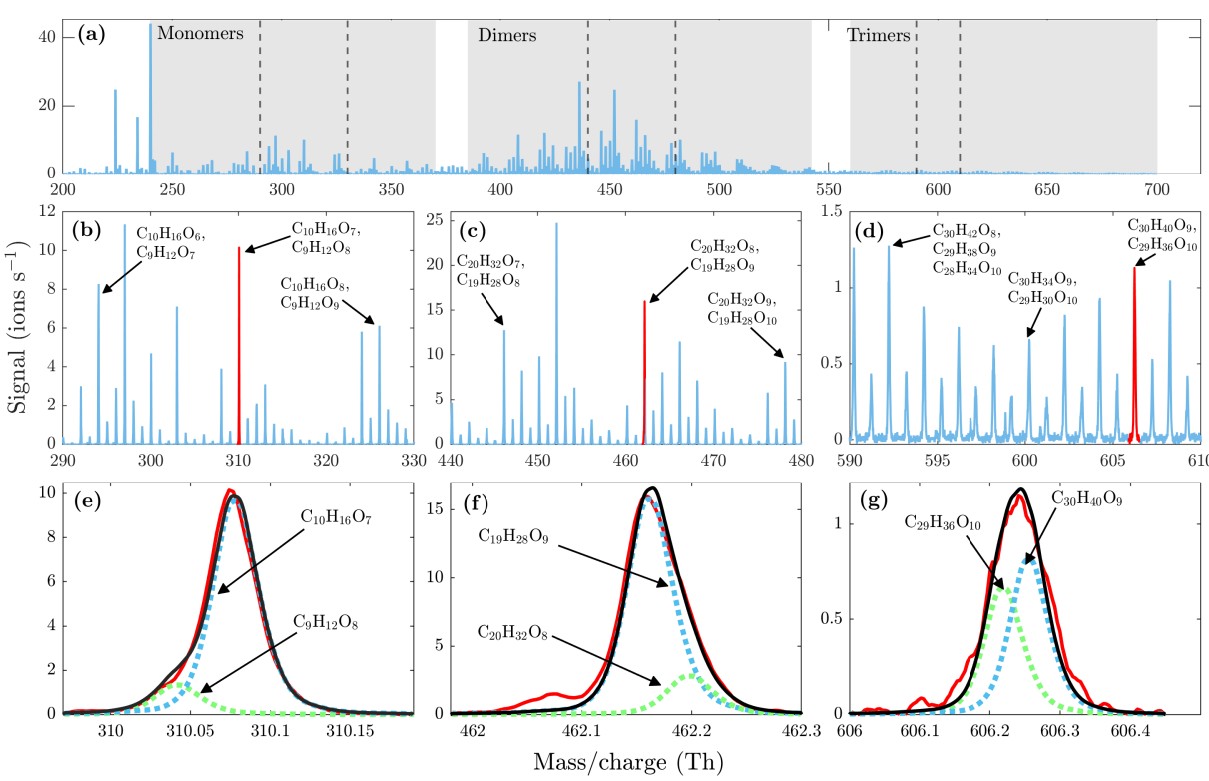

**Figure 6.** (a) High-resolution mass spectrum measured at set temperature of 230 °C at 1 min averaging during $\alpha$-pinene ozonolysis SOA evaporation. Monomer, dimer, and trimer mass ranges are highlighted with grey shading. The dashed lines show the mass ranges that are zoomed in (b), (c) and (d). The high-resolution fits for peaks highlighted with red are shown in (e), (f) and (g). The black lines show the total signal fitted to the measured signals in red. All molecules were detected as clusters with the nitrate ion.





The monomers, dimers, and trimers are also evident in the $NO_3$-CIMS evaporation spectra of other kinds of SOA (Fig. 7),
demonstrating that the VIA coupled to the $NO_3$-CIMS is applicable to different organic species. The full evaporation spectra
for different SOA precursors are included in the Appendix (Fig. A2–A5). We observed that the compounds evaporated from
SOA have less oxygen than the compounds that are typically formed from the precursor oxidation in the gas phase. This
suggests that the particle-phase compounds likely have substantially different functional groups and molecular structures than
the gas-phase compounds. Thermograms for monomers, dimers, and trimers evaporated from different kinds of SOA are shown
in Fig. 8. The thermograms show identical pattern of optimal evaporation temperature increasing with increasing *m/z* ratio until
the signals start to decrease. The decrease of the signals is discussed in more detail in Sect. 4.2. For each type of SOA, dimer
signals reach the highest values, suggesting that SOA contains more dimers than monomers under the loadings and conditions
used in this study. However, previous studies have shown that in the gas phase the monomer concentration is higher than the
dimer concentration (McFiggans et al., 2019; Simon et al., 2020).



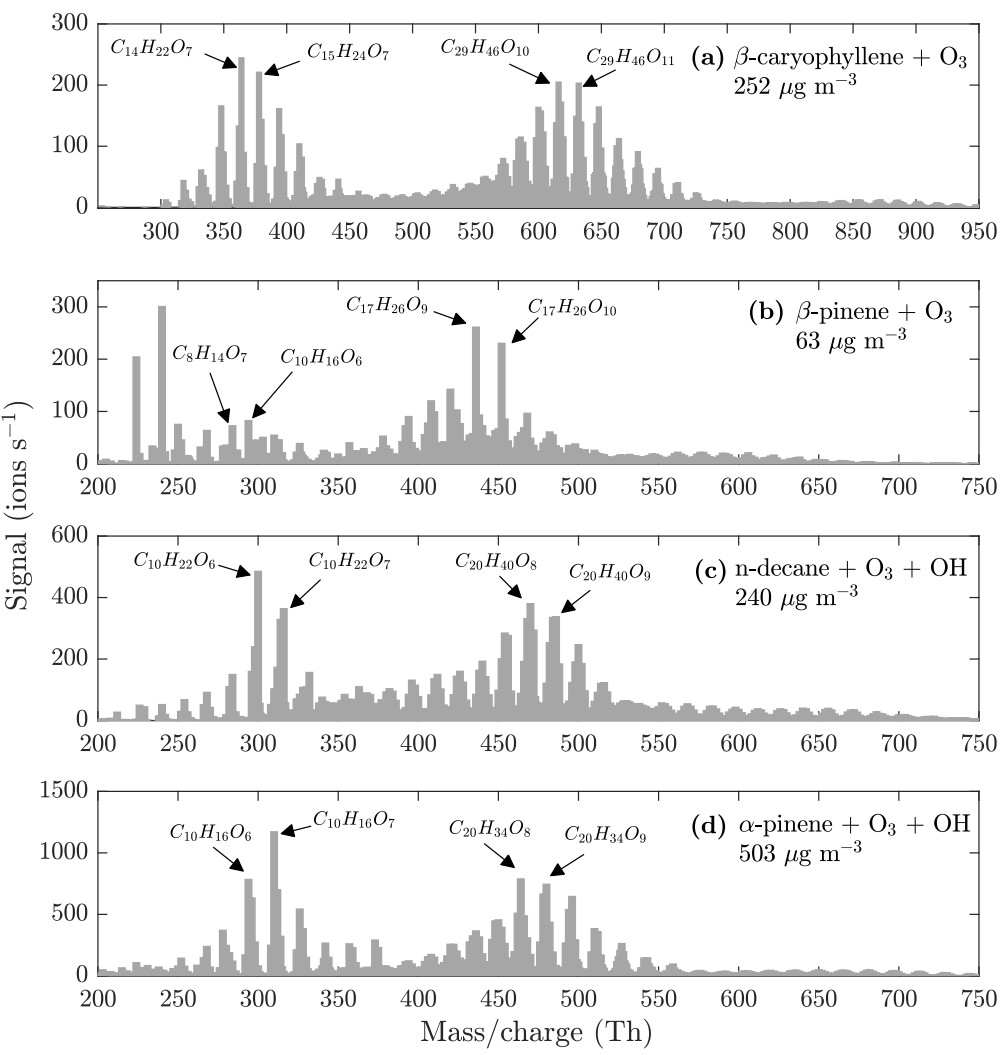

**Figure 7.** Evaporation spectra measured by the NO$_3$-CIMS during the heating of a) $\beta$-caryophyllene SOA b) $\beta$-pinene SOA c) n-decane SOA d) $\alpha$-pinene SOA. The VIA was set to 230 °C. The concentrations are the total aerosol mass loadings in the VIA before dilution. All molecules were detected as clusters with the nitrate ion.





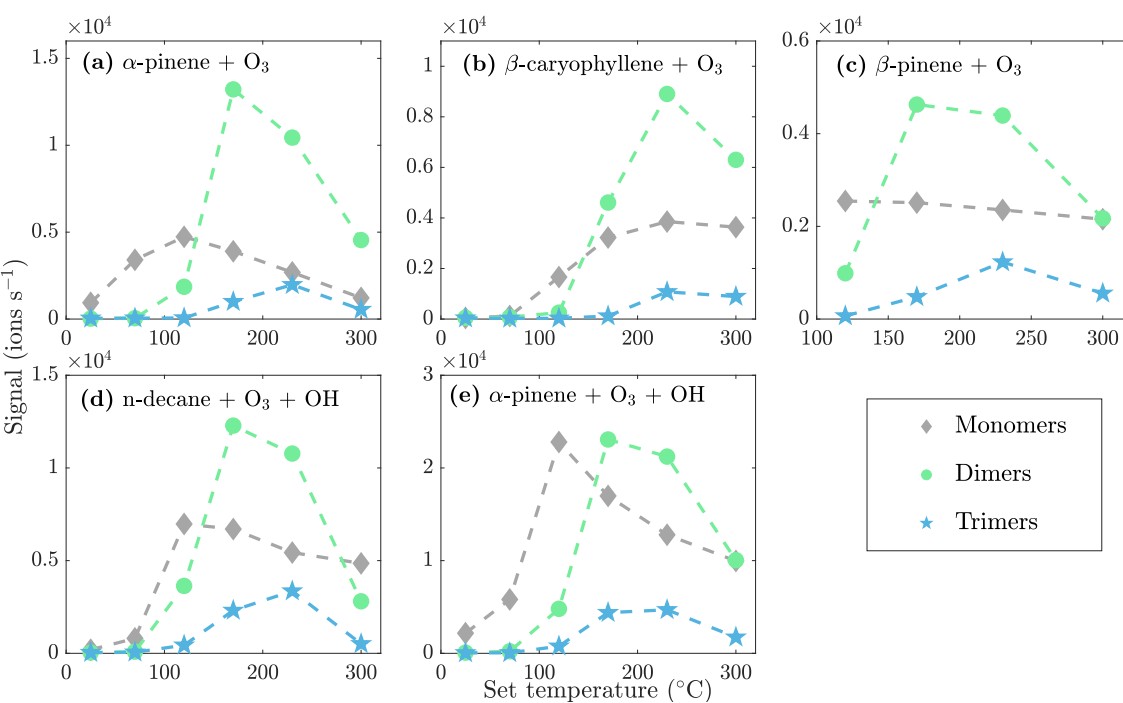

**Figure 8.** Sum of the evaporated monomer, dimer, and trimer signals as a function of the set VIA temperature for different SOA evaporation experiments.





### 230   4.1.1   Comparison of the gas- and particle-phase compounds

To further investigate the differences between the gas and the particle phase, we performed additional experiments in the
environmental chamber. We injected $\alpha$-pinene and ozone into the chamber and measured the gas-phase compounds directly
by the NO$_3$-CIMS. Then, we injected NaCl seed particles with diameter of 80 nm into the chamber to provide surface for the
gas-phase compounds to condense on. We measured the particle-phase compounds using the VIA set to 230 °C and made a
comparison of the a-pinene ozonolysis gas- and particle-phase spectra (Fig. 9a). The sample flow through the VIA was 2 L
min$^{-1}$ and a dilution flow of 8 L min$^{-1}$ of clean air was added after the VIA. Our results are consistent with previous work
(Peräkylä et al., 2020) showing that majority of the compounds with *m/z* ratio larger than 300 Th condense efficiently into
the particle phase due to their low volatilities and few of the compounds with *m/z* ratio smaller than 300 Th are found from
the particle phase. If the gas-phase compounds undergo condensation into the particle phase without a structural change, one
would expect to see a similar signal distribution in the gas and the particle phase, which is not observed here. Instead, the signal
distributions are clearly different in the gas phase compared to the particle phase, suggesting that the condensed compounds are
formed through particle-phase reactions or through some other processes within VIA. Furthermore, molecular level differences
between the gas- and particle-phase compounds (Fig. 9b-d) suggest occurrence of particle-phase reactions, which could lead
to dimer and other larger oligomer formation in the particle phase. We will not estimate the partitioning of the compounds
between the gas and the particle phase, as it is possible that the compounds are transformed after condensing into the particle
phase.

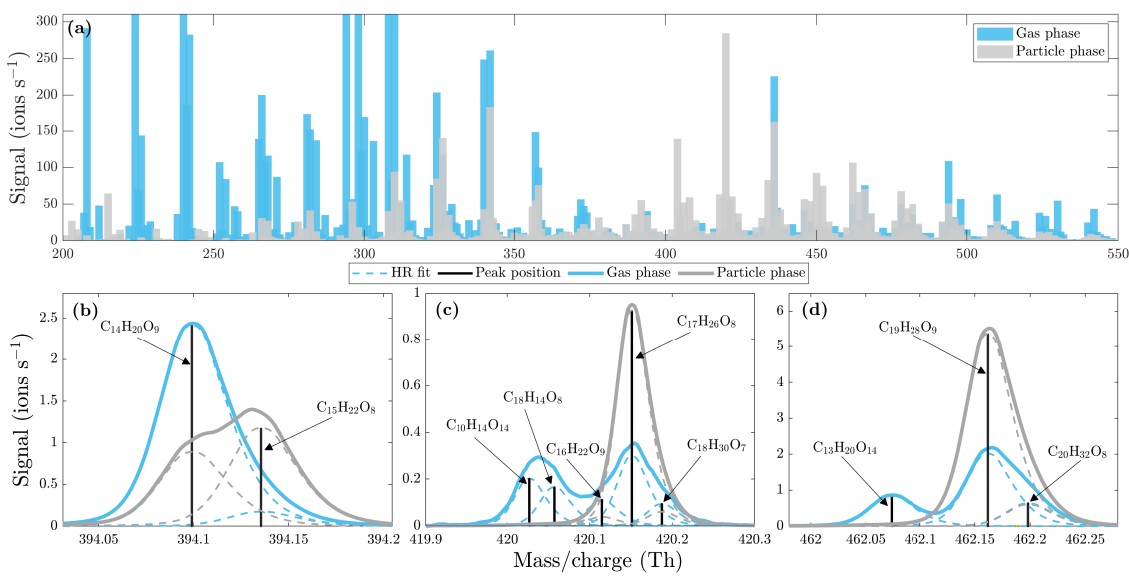

**Figure 9.** Comparison of $\alpha$-pinene ozonolysis gas- and particle-phase spectra (at set temperature of 230 °C) measured by the NO$_3$-CIMS.
a) UMR spectra with 20 min averaging. b), c) and d) high-resolution peak fits. Gas-phase spectrum in blue and particle-phase spectrum in
grey. All molecules were detected as clusters with the nitrate ion.





### 4.1.2 Contribution of highly oxygenated compounds to SOA

Evaporated mass concentration from different kinds of SOA and the atomized sucrose at the different temperature stages is shown in Fig. 10. Concentrations measured by the SMPS are calculated by assuming spherical particles of density of 1400 kg m$^{-3}$ for SOA and of 1600 kg m$^{-3}$ for sucrose. The NO$_3$-CIMS signals were converted to mass concentration using a calibration factor of $2 \times 10^{10}$ cm$^{-3}$ obtained from the ammonium sulfate calibration we performed (see Sect. 3.2). Compounds with *m/z* ratio larger than 200 Th were included in the analysis. As noted in Sect. 4.1, the removal of the gas-phase compounds before the VIA disturbs the gas-particle equilibrium and could lead to evaporation of some organic compounds already at 25 °C. This could explain the larger amount of evaporated mass detected by the NO$_3$-CIMS than the SMPS at 25 °C. Otherwise, the NO$_3$-CIMS detects smaller fraction of the evaporated compounds at lower temperatures compared to higher temperatures. This is expected, as the NO$_3$-CIMS is not able to detect the least oxygenated compounds that evaporate from the particles first. Under the conditions of our experiment, 10 to 20 % of the total SOA mass are highly oxygenated compounds, depending on the SOA precursor (i.e. the volatilities of the corresponding products).

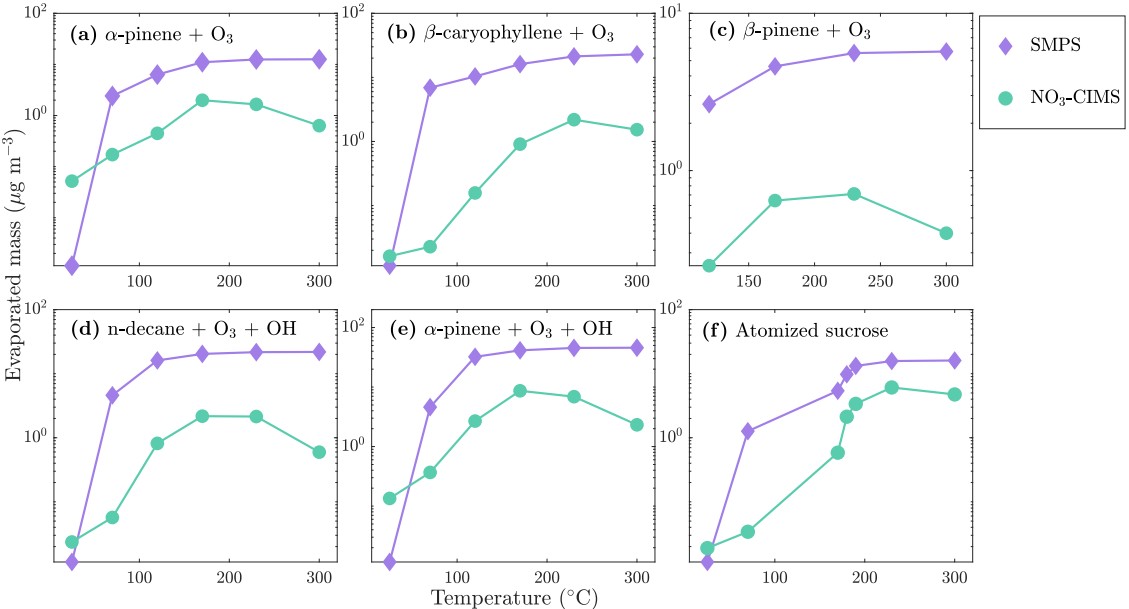

**Figure 10.** Mass evaporated from particles measured by the SMPS and the NO$_3$-CIMS during SOA and sucrose evaporation at different temperatures. A calibration factor of $2 \times 10^{10}$ cm$^{-3}$ used for the NO$_3$-CIMS mass concentration calculations.





## 4.2 Limitations of the method

### 4.2.1 Thermal decomposition

Using thermal evaporation to transfer compounds from particle phase to gas phase has potential limitations, such as thermal decomposition. Thermally unstable large molecules may decompose to form smaller fragments upon heating. These smaller fragments can then be detected, depending on the used gas-phase analyzer. Lopez-Hilfiker et al. (2015) measured acyl-containing compounds from $\alpha$-pinene SOA using FIGAERO and reported thermograms showing secondary modes at much higher tem-

peratures than expected for the specific molecular composition. These secondary modes were most likely caused by larger compounds decomposing to compounds with the same composition as the measured compound. In our results, the decrease in the $NO_3$-CIMS signals after exceeding the optimal evaporation temperature could be attributed to thermal decomposition. When the set temperature was increased from 230 °C to 300 °C, the signals of larger ions (> 500 Th) decreased the most. However, we did not detect considerably large amounts of possible fragments resulting from thermal decomposition of the larger

molecules. The total monomer signal decreased and the signal of a few small ions (< 350 Th) increased. The decomposition fragments are likely to be higher volatility and less oxidized compounds that are not detectable by the $NO_3$-CIMS, and thus not seen in our results. The role of thermal decomposition in this system remains an important topic for future studies.

### 4.2.2 Vapor loss within VIA

Thermal decomposition might not be the only reason for the observed signal decrease. Some compounds might begin to

evaporate nearer the VIA entrance at higher VIA temperatures compared to lower VIA temperatures, and thus have more time to collide with the surrounding walls and be lost. Furthermore, mixing the 1.5 or 2 L min$^{-1}$ sample flow with 8 or 15 L min$^{-1}$ clean air flow after the VIA can lead to turbulence and increased losses of the highly oxygenated compounds between the VIA and the $NO_3$-CIMS. During the ammonium sulfate calibration of the $NO_3$-CIMS, we did not observe signal decrease when the set temperature was increased from 230 °C to 300 °C. However, according to the SMPS data, around 50 % of the

ammonium sulfate mass was evaporated at 230 °C and around 70 % at 300 °C. Due to this, the additional evaporation at 300 °C could compensate the potential signal decrease. For the future experiments, the transmission of the evaporated compounds from VIA to CIMS needs to be improved to minimize the possible losses. Despite these limitations, the $NO_3$-CIMS coupled to the VIA is a promising method able to detect particulate highly oxygenated compounds, even oligomers that are prone to thermal decomposition.

## 5 Conclusions


We present a method for measuring highly oxygenated compounds from organic aerosol using thermal evaporation of aerosol in the VIA followed by identification of the evaporated compounds by the $NO_3$-CIMS. With the experimental set-up used here, we estimate detection limits of below 1 ng m$^{-3}$ for a single compound and below 1 µg m$^{-3}$ for SOA. This can likely be improved by optimization of the set-up, such as increasing the sample flow through the VIA and adding the dilution air





as a sheath flow around the sample flow to maximize the transmission of the evaporated compounds from VIA to CIMS. We demonstrate the detection of particulate highly oxygenated compounds by applying the described method to SOA generated from different precursors. Furthermore, we compare the gas- and particle-phase compounds and see indications of oligomeric compounds formed through particle-phase reactions within SOA.

The VIA allows continuous online aerosol sampling, allowing measurements of fast atmospheric processes. By coupling the

$NO_3$-CIMS to the VIA, we can investigate the transformations and fate of the highly oxygenated compounds after condensing into the particle phase with the same technique used to measure them in the gas phase. Furthermore, the VIA can be coupled to different gas-phase analyzers in order to gain information of the particulate compounds with different volatilities. Further work investigating the potential particle-phase reactions will increase our understanding of the evolution and fate of organic aerosol.

*Data availability.* Data are available upon request from the corresponding author.



**Appendix A**

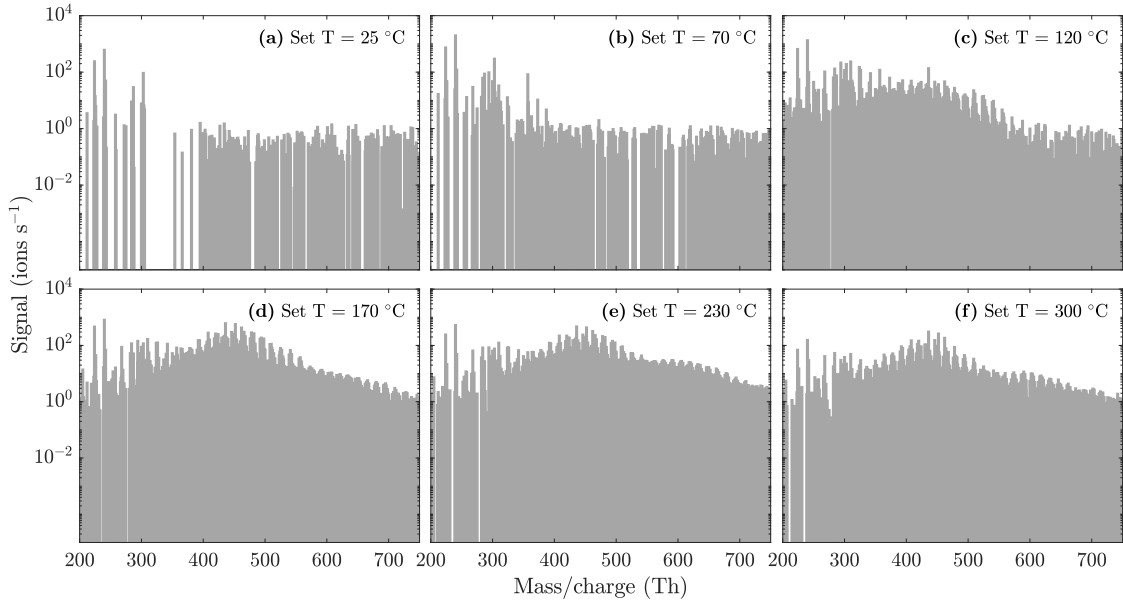

**Figure A1.** NO$_3$-CIMS spectra measured during $\alpha$-pinene ozonolysis SOA evaporation at different temperatures. Same figure as Fig. 5, but with a logarithmic y-scale to better show the trimer region.





**Table A1.** Peak list of the main compounds evaporated from $\alpha$-pinene SOA particles. Note that the mass of the molecules include the nitrate ion.

| Monomers | | Dimers | | Trimers | |
|---|---|---|---|---|---|
| **Composition** | *m/z* | **Composition** | *m/z* | **Composition** | *m/z* |
| $C_7H_{10}O_4$ | 220.0463 | $C_{18}H_{28}O_7$ | 418.1719 | $C_{28}H_{34}O_8$ | 560.2137 |
| $C_5H_6O_6$ | 224.0048 | $C_{17}H_{26}O_8$ | 420.1511 | $C_{29}H_{36}O_9$ | 590.2243 |
| $C_8H_{12}O_4$ | 234.0619 | $C_{19}H_{28}O_7$ | 430.1719 | $C_{30}H_{40}O_8$ | 590.2607 |
| $C_5H_6O_7$ | 239.9997 | $C_{18}H_{28}O_8$ | 434.1668 | $C_{28}H_{34}O_{10}$ | 592.2036 |
| $C_8H_{10}O_5$ | 248.0412 | $C_{17}H_{26}O_9$ | 436.1460 | $C_{29}H_{38}O_9$ | 592.2399 |
| $C_8H_{12}O_5$ | 250.0568 | $C_{19}H_{28}O_8$ | 446.1668 | $C_{28}H_{38}O_{10}$ | 596.2349 |
| $C_7H_{10}O_6$ | 252.0361 | $C_{19}H_{30}O_8$ | 448.1824 | $C_{29}H_{36}O_{10}$ | 606.2192 |
| $C_8H_{12}O_6$ | 266.0518 | $C_{18}H_{28}O_9$ | 450.1617 | $C_{30}H_{40}O_9$ | 606.2556 |
| $C_9H_{12}O_6$ | 278.0518 | $C_{17}H_{26}O_{10}$ | 452.1410 | $C_{29}H_{42}O_{10}$ | 612.2662 |
| $C_{10}H_{16}O_5$ | 278.0881 | $C_{20}H_{30}O_8$ | 460.1824 | $C_{27}H_{40}O_{12}$ | 618.2403 |
| $C_8H_{12}O_7$ | 282.0467 | $C_{19}H_{28}O_9$ | 462.1617 | $C_{28}H_{44}O_{11}$ | 618.2767 |
| $C_8H_{14}O_7$ | 284.0623 | $C_{20}H_{32}O_8$ | 462.1981 | $C_{27}H_{42}O_{12}$ | 620.2560 |
| $C_9H_{12}O_7$ | 294.0467 | $C_{19}H_{30}O_9$ | 464.1773 | $C_{29}H_{36}O_{11}$ | 622.2142 |
| $C_{10}H_{16}O_6$ | 294.0831 | $C_{18}H_{28}O_{10}$ | 466.1566 | $C_{30}H_{40}O_{10}$ | 622.2505 |
| $C_8H_{12}O_8$ | 298.0416 | $C_{17}H_{26}O_{11}$ | 468.1359 | $C_{29}H_{38}O_{11}$ | 624.2298 |
| $C_{10}H_{14}O_7$ | 308.0623 | $C_{20}H_{30}O_9$ | 476.1773 | $C_{29}H_{42}O_{12}$ | 644.2560 |
| $C_{10}H_{16}O_7$ | 310.0780 | $C_{20}H_{32}O_9$ | 478.1930 | $C_{29}H_{44}O_{12}$ | 646.2716 |
| $C_{10}H_{14}O_8$ | 324.0572 | $C_{18}H_{28}O_{11}$ | 482.1515 | $C_{30}H_{46}O_{12}$ | 660.2873 |
| $C_{10}H_{16}O_8$ | 326.0729 | $C_{20}H_{30}O_{10}$ | 492.1723 | $C_{30}H_{44}O_{13}$ | 674.2666 |
| $C_{10}H_{14}O_9$ | 340.0521 | $C_{19}H_{28}O_{11}$ | 494.1515 | $C_{30}H_{46}O_{13}$ | 676.2822 |
| $C_{10}H_{16}O_9$ | 342.0678 | $C_{20}H_{32}O_{10}$ | 494.1879 | $C_{29}H_{44}O_{14}$ | 678.2615 |
| $C_{10}H_{16}O_{10}$ | 358.0627 | $C_{20}H_{32}O_{11}$ | 510.1828 | $C_{30}H_{48}O_{13}$ | 678.2979 |





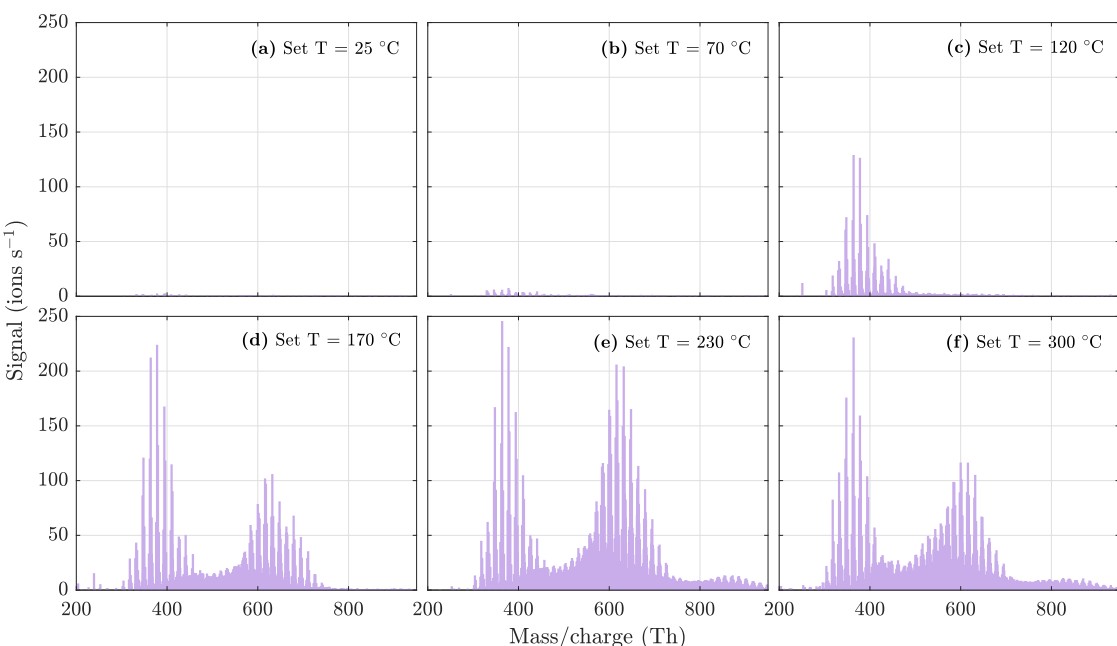

**Figure A2.** NO$_3$-CIMS spectra measured during $\beta$-caryophyllene ozonolysis SOA evaporation at different temperatures. The sample flow through the VIA was 1.5 L min$^{-1}$ and a dilution flow of 15 L min$^{-1}$ of clean air was added after the VIA. The total SOA mass loading was 252 µg m$^{-3}$ in the VIA before dilution.



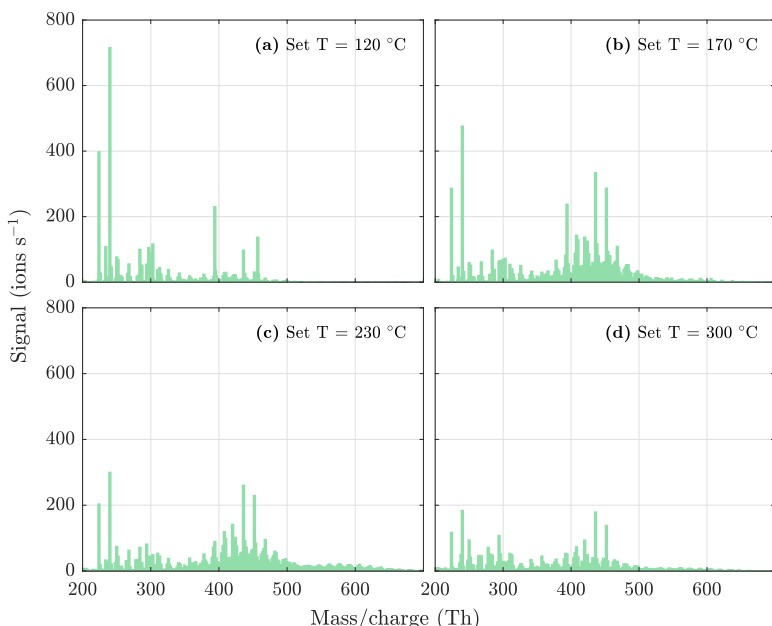

**Figure A3.** NO$_3$-CIMS spectra measured during $\beta$-pinene ozonolysis SOA evaporation at different temperatures. The sample flow through the VIA was 1.5 L min$^{-1}$ and a dilution flow of 15 L min$^{-1}$ of clean air was added after the VIA. The total SOA mass loading was 63 µg m$^{-3}$ in the VIA before dilution.



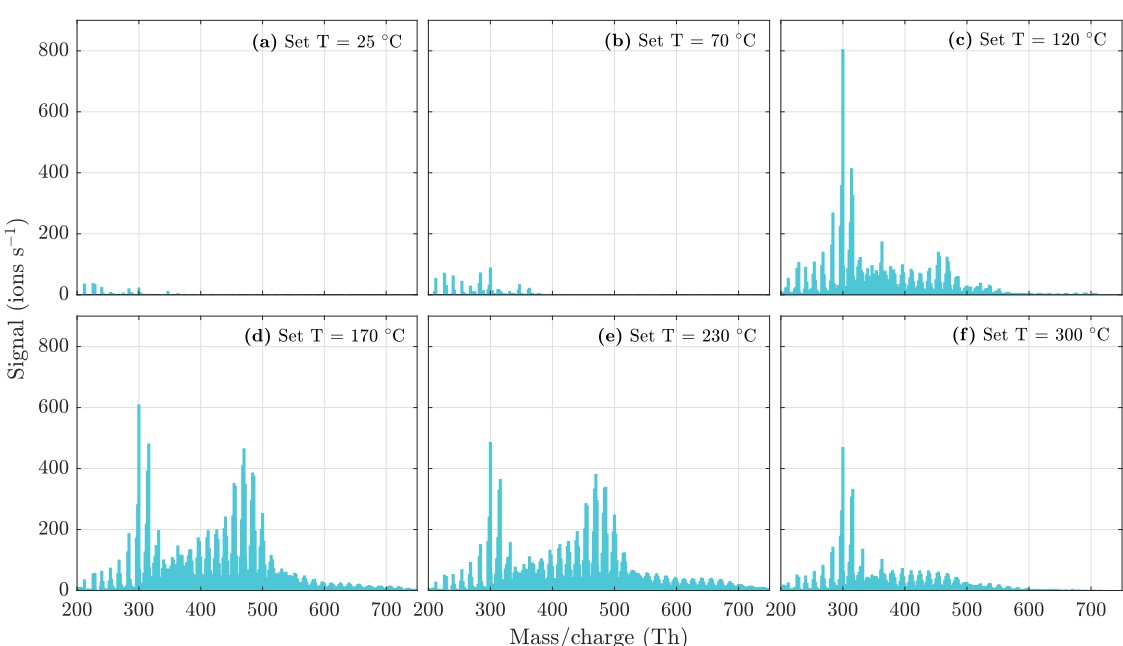

**Figure A4.** NO$_3$-CIMS spectra measured during n-decane ozonolysis and OH oxidation SOA evaporation at different temperatures. The sample flow through the VIA was 1.5 L min$^{-1}$ and a dilution flow of 15 L min$^{-1}$ of clean air was added after the VIA. The total SOA mass loading was 240 µg m$^{-3}$ in the VIA before dilution.





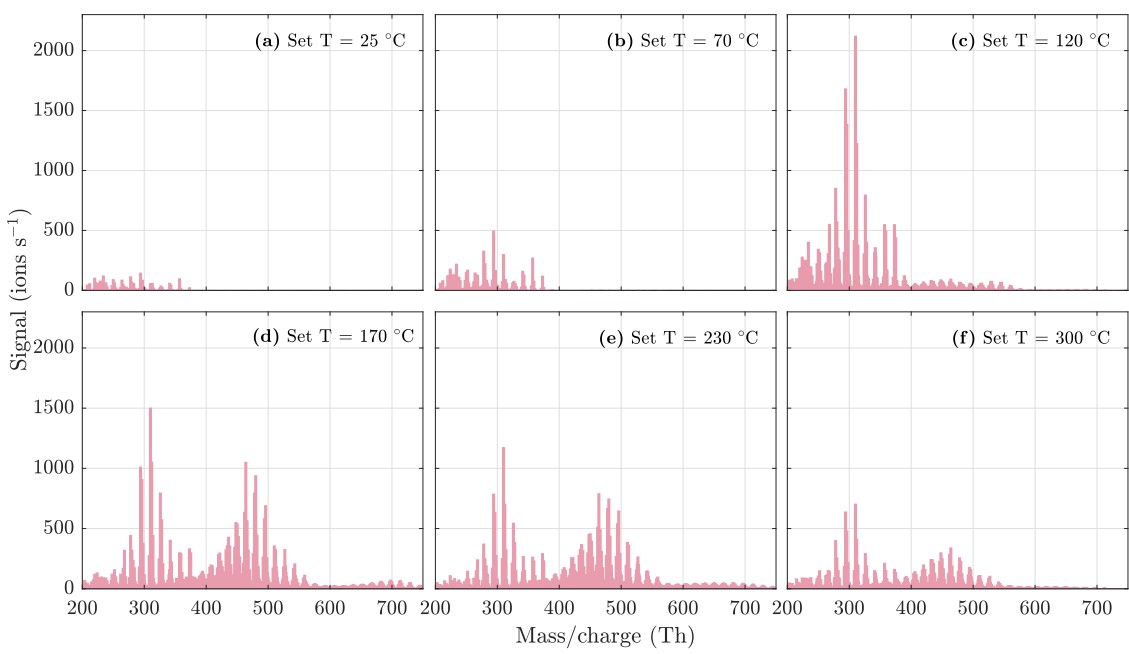

**Figure A5.** NO₃-CIMS spectra measured during α-pinene ozonolysis and OH oxidation SOA evaporation at different temperatures. The sample flow through the VIA was $1.5 \ \mathrm{L \ min^{-1}}$ and a dilution flow of $15 \ \mathrm{L \ min^{-1}}$ of clean air was added after the VIA. The total SOA mass loading was $503 \ \mathrm{\mu g \ m^{-3}}$ in the VIA before dilution.



*Author contributions.* ME and JK designed the study. EH, JZ, FG and NF conducted the experiments. EH and JZ analyzed the data. EH prepared the manuscript with contributions from all co-authors.

*Competing interests.* Jordan Krechmer and Douglas Worsnop work for Aerodyne Research, Inc., which developed the Vocus inlet for aerosols (VIA) used in this study.

*Acknowledgements.* This research was funded by the Academy of Finland (grant nos. 317380, 320094, 325656, 345982 and 346370) and University of Helsinki 3-year grant (75284132). Ella Häkkinen thanks the Vilho, Yrjö and Kalle Väisälä Foundation for financial support. Frans Graeffe thanks Svenska Kulturfonden for financial support. Nicolas Fauré has been supported by the Swedish Research Councils, VR (2020-03497).



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
