# Peer review of "Online measurement of highly oxygenated compounds from organic aerosol"

_EGUsphere, 2022_

## Author Comment (AC1)

**Response to Reviewer #1**

**General comment:**

The authors characterize a new Vocus Inlet for Aerosols (VIA) inlet, which thermally evaporates particles for online sampling. Coupled with an NO3-CIMS, they demonstrate the use of the VIA for measuring the chemical composition of several different aerosol types. These measurements show that the VIA could be useful for making online speciated particle composition measurements. The authors also discuss several artifacts and possible unknowns that need to be further investigated before the VIA can really be trusted to provide accurate speciation of highly oxygenated organic compounds. It would be best if the authors could provide direct comparisons of speciated VIA-NO3CIMS measurements with one or more of the other particle phase measurements discussed in the introduction (see my last comment below), which could give the reader a better idea of how well this method works in comparison to other methods, but that may not have been feasible. After addressing my comments below, I recommend for publication.

We thank the reviewer for the positive feedback and answer the comments point-by-point below:

**Comment #1:**

Line 25 and throughout: Why have you chosen to use the term "highly oxygenated compounds" throughout the manuscript instead of HOMs? Is there a technical difference between highly oxygenated compounds and HOMs that you can give in the text to clarify this, or can you instead refer to them as HOMs everywhere?

**Response:**

Our reason for not using the term "HOM" was that many of the compounds we observe do not match with the definitions recommended by Bianchi et al., 2019 (e.g. formed via autoxidation in the gas phase and typically containing six or more oxygen atoms). In our experiments, we detect compounds containing four or more oxygen atoms in the particle phase, and we are not able to determine anything about their formation pathways. Nevertheless, this comment made us realize that we do not adequately describe this choice in the manuscript, and the similarity of the terms may lead to confusion. We added a motivation to the methods (Sect. 2.2.1) for this choice, and defined better our terminology as follows: "The nitrate-ion-based chemical ionization mass spectrometer ($NO_3CIMS$) is selective towards low-volatile highly oxygenated compounds, and therefore we chose to use it in this study to detect the evaporated compounds from the particle phase. Importantly, the term "HOM" (highly oxygenated organic molecules) has been used in many studies to describe the group of compounds measured by the $NO_3$-CIMS, and the recommended definition for this term, as proposed by Bianchi et al, (2019) is based on gas-phase measurements. While many of the compounds we detect with the VIA are similar to the HOM, there are enough differences that we opted to not use the term "HOM" to describe the compounds detected here. In particular, the criteria relating to compounds having been formed via autoxidation in the gas phase and typically containing six or more oxygen atoms, are not valid (or it cannot be verified) for all observed compounds in our study, as will be shown later. We will still describe the compounds as highly oxygenated, but in practice we will be referring to all compounds that we were able to detect with this new set-up."

**Comment #2:**

Line 32: I'm not sure what you mean by "form larger components of SOA". I think you mean form larger molecules through oligomerization, but this could also mean they just accumulate until they comprise a large component of the SOA by mass. Please clarify.

**Response:**

To clarify this, we modified the sentence as follows: "It has been suggested that after condensation, HOM either remain in the particle phase without structural changes, undergo fragmentation reactions, or undergo accretion reactions (Mutzel et al., 2015; Bianchi et al., 2019)."

**Comment #3:**

Line 136: This paragraph is a bit confusing. First you say the internal temperature is not known, then you say you measured it and it internally reaches the setpoint by the outlet, then you go back to using the external temps and saying you use the setpoints and they aren't necessarily the same as actual internal temps. I think it would be best if you show/state the internal measurements, but it would also be fine to just replace this paragraph with a simple sentence saying the temps are controlled externally so the internal temps may vary slightly.

**Response:**

We added a figure of the measured temperature profile to the appendix of the manuscript, and revised the paragraph as follows: "The VIA temperature is set with respect to the measured temperature values from the outer surface of the thermal desorption unit. To investigate the temperature profile inside the thermal desorption unit, we placed a thermocouple into different locations inside the unit and measured the temperature of the gas at different set temperatures and flow rates (Fig. A1). The temperature profile inside the unit was not uniform, as the gas was colder at the entrance than at the outlet of the unit. The temperatures reported in this study are the setpoint values of the thermal desorption unit of the VIA."

**Comment #4:**

Line 160: By "decreasing trend", do you mean the fact that the linear regression slopes are less than 1? This language could probably be clarified. But my main question here is what role could wall losses of H2SO4 or highly oxidized organics to the tubing between the VIA and the NO3-CIMS be playing? Could that be causing this effect at higher concentrations?

**Response:**

The reviewer is correct, and this was poorly chosen wording. We indeed tried to indicate the less steep increase of the measured concentrations by the $NO_3$-CIMS as compared to the SMPS. The suggested wall losses are indeed also expected to be important, but to a first approximation, the fraction of molecules lost between the VIA and the $NO_3$-CIMS should be constant, and therefore we do not expect that this would be a reason for the different slopes. We clarified the text on line 160

as below. We also refer to our response to the comment #2 from the reviewer 2 concerning the calibration for some additional discussion that may be relevant to this topic.

We revised the paragraph as follows: "The calibration curves are not as steep as the 1:1 line. This could be due to nucleation under high concentration of sulfuric acid in the tubing between the VIA and the $NO_3$-CIMS. Nucleation would lead to a decrease in the gaseous sulfuric acid, resulting in lower mass concentration detected by the $NO_3$-CIMS compared to the SMPS (which simply measures the evaporation from the larger particles) as a function of the evaporated sulfuric acid. Wall losses between the VIA and the $NO_3$-CIMS will certainly also take place, but this should to a first approximation be a constant factor, and as such, becomes included in the calibration factor (Jokinen et al., 2012). On average, the difference between the concentrations measured by the SMPS and the $NO_3$-CIMS is smallest when a calibration factor of $2\times10^{10}$ $cm^{-3}$ is used. Therefore, we decided to use the calibration factor of $2\times10^{10}$ $cm^{-3}$ when converting the $NO_3$-CIMS signals to concentrations. This value is close to the literature calibration factor values that range between $1.1\times10^{10}$ and $1.89\times10^{10}$ (Jokinen et al., 2012; Kürten et al., 2012; Ehn et al., 2014). We acknowledge that our estimate comes with a very large uncertainty, and thus provides concentrations of highly oxygenated compounds with large uncertainties. Hence, we focus more on the qualitative than on the quantitative analysis of the measured data."

**Comment #5:**

Line 163: How does this calibration value of 2x10^10 cm^-3 compare to calibrations previously determined for NO3-CIMS instruments? Is it perhaps lower due to sampling line losses between the VIA and the NO3-CIMS in this setup? Please comment on this.

**Response:**

We now added some the literature values to the text, showing that the value is at the upper end of the reported values (a higher value means lower sensitivity, e.g. from increased sampling losses). We also emphasize that the calibration factor always implicitly includes the losses in the sampling lines. For example, if making a longer sampling line and the extension has a transmission of 50% for molecules of interest, the calibration factor will become twice as large. We also refer to our response to the comment #2 from the reviewer 2 concerning the literature values.

**Comment #6:**

Line 173: Can you apply your calibration factor to the organic compounds here to give a rough estimate of mass concentration measured by the NO3-CIMS?

**Response:**

We applied the calibration factor to the organic compounds and revised the text as follows: "When the SOA mass loading in the chamber was below 0.5 µg $m^{-3}$, the $NO_3$-CIMS signals for compounds 200-750 $m/z$ had started to increase. Therefore, with the crude sampling set-up used here, we found a detection limit of well below 1 µg $m^{-3}$ for organic aerosol. By applying calibration factor of $2\times10^{10}$ $cm^{-3}$, we estimate that the mass concentration of the 200-750 $m/z$ organic compounds was 0.15 µg $m^{-3}$ when the total SOA mass was 0.5 µg $m^{-3}$."

**Comment #7:**

Line 174: The detection limit will depend on the dilution amount relative to sample flow, right? Since you're diluting here by a factor of 4 (2 lpm sample flow and 8 lpm dilution flow), does that mean the limit of detection should be stated as 0.25 ug m-3 mass concentration at the NO3-CIMS inlet?

**Response:**

This is correct to the extent that if we for example had five VIA systems running in parallel, the detection limit could become lower, as suggested. However, reporting this type of value would be misleading for two reasons. Firstly, the dilution also serves the purpose of cooling the flow before the CIMS. Secondly, the detection limit of the $NO_3$-CIMS in general is below $10^5$ $cm^{-3}$ for a pure species, which converts to a detection limit of below 1 $ng/m^3$ in mass concentration. In other words, the detection limit is primarily governed by the VIA setup itself, in addition to the chemical selectivity of the $NO_3^-$ ionization.

**Comment #8:**

Line 222: Does this sentence refer to the data in Fig. 9 and the discussion in Sect. 4.1.1? If so, I suggest removing this statement from here. You are already discussing this concept of comparing the oxygen number between gas and particle measurements in 4.1.1. If you are actually referring to something else in this sentence, please provide a citation or show data to back up this assertion.

**Response:**

We removed the statement "we observed that the compounds evaporated from SOA have less oxygen than the compounds that are typically formed from the precursor oxidation in the gas phase. This suggests that the particle-phase compounds likely have substantially different functional groups and molecular structures than the gas-phase compounds.", as the concept is discussed in detail in the following section (Sect. 4.1.1).

**Comment #9:**

Line 243: Is there any literature on how fast these particle phase reactions could be at room temperature? That could help inform whether these particle phase reactions are occurring before the particles enter the VIA, or if the heat in the VIA is the real cause of particle phase changes. If the latter, that's not so great for trying to identify specific compounds in the particle phase.

**Response:**

Very few studies have been able to measure very short particle phase-reaction times, and Krapf et al., 2016 have reported the lifetime of organic peroxides in α-pinene SOA to be less than an hour at room temperature. This is a very valid question from the reviewer, and we aim to do more systematic studies where we vary the age of the sampled particles to deduce exactly what is happening in the particles and what might be happening upon heating.

**Comment #10:**

General comment: I'm wondering if you have tried sampling the gas phase (without seed aerosol) through the VIA with heating, to compare with normal gas phase sampling without heating. Does the heat lead to gas phase reactions?

**Response:**

Unfortunately, we have not sampled the gas-phase compounds through the VIA system, in part because the compounds of interest that the $NO_3$-CIMS can detect are efficiently lost to surfaces, and the low flow rate through will lead to extremely high losses. In addition, the particle would need to be removed from the sample, but any particle filter will also remove the highly oxygenated compounds. As such, this type of experiment measuring the response of only gas-phase compounds to elevated temperature would require a quite different setup than what we had available.

**Comment #11:**

Line 257: Can you compare this 10-20% number with previous literature or expectations?

**Response:**

These numbers are in the ballpark of what we might have expected, but there are no literature values that we could directly compare to, given that the $NO_3$-CIMS has a very specific selectivity towards certain organic molecules, and particle phase measurements with this approach are extremely limited. Comparison to gas phase data is also complicated, as we also show that the molecules detected in the particles did not resemble the typically detected gas phase HOM spectra. Finally, the large uncertainty in our quantification means that the correct number can easily be outside this "best guess" range.

We now added values from a few previous studies and our expectations to the text as follows: "Under the conditions of our experiment, our best estimate is that 10 to 20 % of the total SOA mass were highly oxygenated compounds that the $NO_3$-CIMS was able to detect. Previous studies of α-pinene ozonolysis have estimated that the contribution of condensing HOM to total SOA mass was 50 to 70 % at lower SOA loadings (~10 µg/m3) than what we used in the current study (Ehn et al., 2014, Roldin et al., 2019). For SOA loadings as high as we used, we expect that the contribution of highly oxygenated compounds to total SOA mass would decrease due to partitioning of more volatile and less oxygenated species to the particles. Given this effect, the unknown particle-phase processes that can alter the SOA composition, and the large uncertainty in our quantification, our estimate seems to be within a reasonable and plausible range. More exact numbers will require further work, to both improve the sampling and the quantification of our new VIA $NO_3$-CIMS design."

**Comment #12:**

Line 271: Could the thermal decomposition be not just breaking C=C bonds, but loss of water, CO, CO2, etc leaving you with less oxygenated species that are no longer clustering with NO3-? I'd suggest adding a sentence discussing this here.

**Response:**

To clarify that we use the term "decomposition fragments" to describe any kind of fragmentation or decomposition that can lead to the $NO_3$-CIMS not detecting the initial molecule or/and either of the products (including the cases the reviewer pointed out), we revised the text as follows: "The products of the decomposition are likely to be higher volatility and less oxygenated compounds that cannot be detected by the $NO_3$-CIMS, and thus not seen in our results."

**Comment #13:**

Another general comment: In the introduction, you summarized well the existing particle phase speciation techniques. I am wondering if there are opportunities to directly compare measurements taken with the VIA and NO3-CIMS to some of these other techniques. For example, are there any published spectra of e.g. alpha pinene ozonolysis SOA that measure similar compounds, that you could show a direct comparison with? The ionization techniques might be different, but it would be interesting to see such a comparison. Sampling the same aerosol with multiple techniques would of course be the best way to directly compare, but may not be feasible depending on the authors' access to the other techniques.

**Response:**

It would indeed be great to do a direct comparison by sampling the same aerosol with multiple techniques. Unfortunately, we did not have the chance, but we hope that we could do it soon in the future. It is difficult to compare spectra measured with nitrate ionization and some other ionization technique, as the nitrate does not seem to detect overlapping species with any other ionization technique, e.g. iodide that is commonly used with FIGAERO (Riva et al., 2019). However, in our follow-up publication using VIA and $NO_3$-CIMS we did a comparison on the particle-phase HOM measured by VIA-$NO_3$-CIMS and other techniques (Zhao et al, 2022).

**References:**

Bianchi, F., Kurten, T., Riva, M., Mohr, C., Rissanen, M. P., Roldin, P., 310 Berndt, T., Crounse, J. D., Wennberg, P. O., Mentel, T. F., Wildt, J., Junninen, H., Jokinen, T., Kulmala, M., Worsnop, D. R., Thornton, J. A., Donahue, N., Kjaergaard, H. G., and Ehn, M.: Highly Oxygenated Organic Molecules (HOM) from Gas-Phase Autoxidation Involving Peroxy Radicals: A Key Contributor to Atmospheric Aerosol, Chemical Reviews, 119, 3472–3509, https://doi.org/10.1021/acs.chemrev.8b00395, 2019.

Ehn, M., Thornton, J. A., Kleist, E., Sipila, M., Junninen, H., Pullinen, I., Springer, M., Rubach, F., Tillmann, R., Lee, B., Lopez-Hilfiker, F., Andres, S., Acir, I. H., Rissanen, M., Jokinen, T., Schobesberger, S., Kangasluoma, J., Kontkanen, J., Nieminen, T., Kurten, T., Nielsen, L. B., Jorgensen, S., Kjaergaard, H. G., Canagaratna, M., Maso, M. D., Berndt, T., Petaja, T., Wahner, A., Kerminen, V. M., Kulmala, M., Worsnop, D. R., Wildt, J., and Mentel, T. F.: A large source of low-volatility secondary organic aerosol, Nature, 506, 476-479, 10.1038/nature13032, 2014

Krapf, M., El Haddad, I., Bruns, Emily A., Molteni, U., Daellenbach, Kaspar R., Prévôt, André S. H., Baltensperger, U., and Dommen, J.: Labile Peroxides in Secondary Organic Aerosol, Chem, 1, 603-616, https://doi.org/10.1016/j.chempr.2016.09.007, 2016.

Mutzel, A., Poulain, L., Berndt, T., Iinuma, Y., Rodigast, M., Böge, O., Richters, S., Spindler, G., Sipilä, M., Jokinen, T., Kulmala, M., and Herrmann, H.: Highly Oxidized Multifunctional Organic Compounds Observed in Tropospheric Particles: A Field and Laboratory Study, Environmental Science & Technology, 49, 7754–7761, https://doi.org/10.1021/acs.est.5b00885, 2015.

Riva, M., Rantala, P., Krechmer, J. E., Peräkylä, O., Zhang, Y., Heikkinen, L., Garmash, O., Yan, C., Kulmala, M., Worsnop, D., and Ehn, M.: Evaluating the performance of five different chemical ionization techniques for detecting gaseous oxygenated organic species, Atmos. Meas. Tech., 12, 2403-2421, doi:10.5194/amt-12-2403-2019, 2019.

Zhao, J., Häkkinen, E., Graeffe, F., Krechmer, J. E., Canagaratna, M. R., Worsnop, D. R., Kangasluoma, J., and Ehn, M.: A Combined Gas- and Particle-phase Analysis of Highly Oxygenated Organic Molecules (HOM) from α-pinene Ozonolysis, EGUsphere, 2022, 1-30, 10.5194/egusphere-2022-1317, 2022.

---

## Author Comment (AC2)

**Response to Reviewer #2**

**General comment:**

In this manuscript rather new technique - Vocus Inlet for Aerosols (VIA) is briefly described and coupled with NO3-CIMS applied for measurement and characterization of organic aerosol. VIA is traditionally coupled with Vocus PTR, therefore this study represents an important extension of VIA's usability in atmospheric science by showing successful real time measurement of complex composition of secondary aerosol with impressive detection limits. Authors have put significant thought into the experimental set up, background corrections and themselves highlight the pros and cons of such technique for reliable aerosol measurement. It would be interesting to see such measurement in comparison to techniques targeting similar goal such as EESI or Figaero. As first attempt to make aerosol measurements with this coupling I find the study well done and recommend for publication after addressing minor comments.

We thank the reviewer for the positive feedback and answer the comments point-by-point below:

**Comment #1:**

135: The temperature profile inside the unit was not uniform'' - Could authors provide the numbers of these measurements? I assume 300C is the maximum temperature VIA can reach and at recommended flow not sufficient for total evaporation as authors state. Or would this be compensated through losses and decreased particle transmission. The particle transmission was measured at 1 l/min however 1.5 l/min or 2 l/min were used throughout the experiment without clarification why. Again, could authors provide results of the size dependance measurement.

**Response:**

We added a figure on the measured temperature profile to the appendix of the manuscript (Fig. A1). The reviewer is correct that the maximum temperature of VIA is roughly 300 °C and that 70 % of the mass of ammonium sulfate was evaporated at this temperature. However, for SOA, almost all (> 99%) of the mass evaporated at 300 °C. As particle losses decrease with higher flow rates, we measured the particle transmission at 1 l/min to provide the lower limit. Unfortunately, we have not measured the size dependence, but we will do it soon in the future.

**Comment #2:**

160: The way the calibration is presented is a bit confusing. Shall the reader understand that three time-separate calibrations were made corresponding to presented calibration factors of 1- 3 x10^10 cm^-3? It is not clear that these calibration factors originate from separate measurements. How do these calibration factors compare to literature? How do authors explain the non linear step between 0.8 – 1 ng/m3 for all the curves? Could losses explain the discrepancy in the calibrations? Did authors calibrate higher than 1 ng/m3? Please clarify.

**Response:**

The calibration factors do not originate from separate measurements. We performed the experiment with the purpose of determining the instrument sensitivity, i.e. the calibration factor. In Figure 2, we simply converted the signals measured by the $NO_3$-CIMS to mass concentrations by applying three different calibration factors that make the results between the instruments match. It turned out that these factors were close to values reported earlier in literature (Jokinen et al., 2012; Kürten et al., 2012; Ehn et al., 2014). The nonlinear step could be due to enhanced nucleation under a high concentration of sulfuric acid, but it can also simply be uncertainty in the measurements. We emphasize that the three curves describe the same experiment, and thus are not independent from each other, and therefore there is no clear systematic trend to be seen. To limit potential sulfuric acid nucleation affecting our results, we did not use higher mass concentrations than 1 ng/m$^3$.

We modified the text discussing Fig. 2 as follows: "The calibration curves are not as steep as the 1:1 line. This could be due to nucleation under a high concentration of sulfuric acid in the tubing between the VIA and the $NO_3$-CIMS. Nucleation would lead to a decrease in the gaseous sulfuric acid, resulting in lower mass concentration detected by the $NO_3$-CIMS compared to the SMPS (which simply measures the evaporation from the larger particles) as a function of the evaporated sulfuric acid. Wall losses between the VIA and the $NO_3$-CIMS will certainly also take place, but this should to a first approximation be a constant factor, and as such, becomes included in the calibration factor (Jokinen et al., 2012). On average, the difference between the concentrations measured by the SMPS and the $NO_3$-CIMS is smallest when a calibration factor of $2\times10^{10}$ cm$^{-3}$ is used. Therefore, we decided to use the calibration factor of $2\times10^{10}$ cm$^{-3}$ when converting the $NO_3$-CIMS signals to concentrations. This value is close to the literature calibration factor values that range between $1.1\times10^{10}$ and $1.89\times10^{10}$ (Jokinen et al., 2012; Kürten et al., 2012; Ehn et al., 2014). We acknowledge that our estimate comes with a very large uncertainty, and thus provides concentrations of highly oxygenated compounds with large uncertainties. Hence, we focus more on the qualitative than on the quantitative analysis of the measured data."

**Comment #3:**

165: For readers not having experience with Nitrate Cims, please explain the unit for the calibration factor.

**Response:**

To clarify this, we modified Sec. 2.2.1 as follows: "The signal intensity of highly oxygenated compound X can be converted to a concentration by normalizing the measured signal intensity (ions s$^{-1}$) with the total measured reagent ion signal (ions s$^{-1}$), which cancels out the unit. Then the normalized value is multiplied by the calibration factor (cm$^{-3}$), yielding the concentration of X in units cm$^{-3}$, which describes the number of molecules of X in a cubic centimeter of gas (Eq. (2), Jokinen et al., 2012)."

**Comment #4:**

170: What role does the gas-particle equilibrium play? This equilibrium is distorted 1. Via the denuder and 2. via the dilution with clean air on potentially still present particles. Would this somehow effect your results? Or is this effect negligible?

**Response:**

The reviewer is correct that the denuder and dilution can disturb the gas-particle equilibrium. However, since VOC oxidation will be on-going up until the VIA denuder, the gases and particles will not be in true equilibrium before the denuder either. As an example, almost all typically detected gas-phase HOM are only detected because they have not had enough time to condense (i.e. the partitioning equilibrium is heavily on the particle side). Nevertheless, the residence times are so short (seconds or less) that we indeed expect the effects to be every small. For some SVOC, which the $NO_3$-CIMS typically is not very good at detecting, the effects should be the largest. As discussed in Sec. 4.1, we detected some organic compounds of higher volatility (mainly C5 compounds) evaporating when VIA was set to 25 °C. This net evaporation is likely a result of the disturbed gas-particle equilibrium, i.e. that the gas phase was removed in the denuder. In addition, at the point of dilution, the particles have already undergone heating, and therefore there should be negligible amounts of compounds left that would be affected by the dilution at room temperature.

**Comment #5:**

175: In this case is the author talking about background (filter) NO3-CIMS corrected signal?

**Response:**

This is correct; the $NO_3$-CIMS background signal is subtracted from the $NO_3$-CIMS signal we show in Fig. 3.

**Comment #6:**

190: With detection limits as low as 1 ug/m3 why did authors choose to investigate SOA at such high loadings?

**Response:**

We wanted to generate high enough loadings of SOA to ensure that we measure well above the detection limit, as we determined the exact value of the detection limit after the experiments.

**Comment #7:**

215: There has been studies showing that the prevalence of oligomers is linked to total SOA mass. I encourage authors to consider this option and provide references.

**Response:**

The reviewer is correct that higher concentration of precursors (higher SOA loading) may cause higher amount of oligomers formed in the gas or particle phase (Kourtchev et al., 2016). However, in our follow-up publication using VIA and $NO_3$-CIMS, we performed experiments with lower SOA loadings (1-14 µg m$^{-3}$) and still detected oligomers (Zhao et al, 2022).

**Comment #8:**

250: What does prevent the author to use calibration factor for sucrose from direct sucrose calibration? This would be technically possible the same way as for ammonium sulfate and could diminish the discrepancy in the figure at high temperature

**Response:**

In Fig. 10f, we show the evaporated mass from sucrose particles measured by the SMPS and the $NO_3$-CIMS. We included all compounds with *m/z* ratio larger than 200 Th in this analysis and used calibration factor of $2 \times 10^{10}\,cm^{-3}$ to convert the $NO_3$-CIMS signals to concentrations. In order to reach similar mass concentrations than the SMPS measured for sucrose, the $NO_3$-CIMS calibration factor would need to be $\sim 9 \times 10^{10}\,cm^{-3}$. The signal from sucrose particles was spread out over wide *m/z* range (see Fig. 4f), and we detected smaller and larger compounds than sucrose itself evaporating from the particles, potentially due to fragmentation and particle-phase reactions. In addition to this, we cannot be sure that we are sensitive to all the evaporating species from sucrose particles. In the sulfuric acid calibration, the signal of the evaporated sulfuric acid from ammonium sulfate particles was distributed to $H_2SO_4NO_3^-$ and $HSO_4^-$ ions, giving a calibration factor close to the literature values.

**References:**

Ehn, M., Thornton, J., Kleist, E., Sipila, M., Junninen, H., Pullinen, I., Springer, M., Rubach, F., Tillmann, R., Lee, B., Lopez-Hilfiker, F., Andres, S., Acir, I.-H., Rissanen, M., Jokinen, T., Schobesberger, S., Kangasluoma, J., Kontkanen, J., Nieminen, T., Kurten, T., Nielsen, L., Jorgensen, S., Kjaergaard, H., Canagaratna, M., Dal Maso, M., Berndt, T., Petaja, T., Wahner, A., Kerminen, V.-M., Kulmala, M., Worsnop, D., Wildt, J., and Mentel, T.: A large source of low-volatility secondary organic aerosol, Nature, 506, 476–+, https://doi.org/10.1038/nature13032, 2014.

Jokinen, T., Sipila, M., Junninen, H., Ehn, M., Lonn, G., Hakala, J., Petaja, T., Mauldin III, R. L., Kulmala, M., and Worsnop, D. R.:Atmospheric sulphuric acid and neutral cluster measurements using CI-APi-TOF, Atmospheric Chemistry and Physics, 12, 4117–4125, https://doi.org/10.5194/acp-12-4117-2012, 2012.

Kourtchev, I., Giorio, C., Manninen, A., Wilson, E., Mahon, B., Aalto, J., Kajos, M., Venables, D., Ruuskanen, T., Levula, J., Loponen, M., Connors, S., Harris, N., Zhao, D., Kiendler-Scharr, A., Mentel, T., Rudich, Y., Hallquist, M., Doussin, J.-F., Maenhaut, W., Bäck, J., Petäjä, T., Wenger, J., Kulmala, M., and Kalberer, M.: Enhanced Volatile Organic Compounds emissions and organic aerosol mass increase the oligomer content of atmospheric aerosols, Sci. Rep., 6, 35038, 10.1038/srep35038, 2016.

Kürten, A., Rondo, L., Ehrhart, S., and Curtius, J.: Calibration of a Chemical Ionization Mass Spectrometer for the Measurement of Gaseous Sulfuric Acid, The Journal of Physical Chemistry A, 116, 6375–6386, https://doi.org/10.1021/jp212123n, 2012.

Zhao, J., Häkkinen, E., Graeffe, F., Krechmer, J. E., Canagaratna, M. R., Worsnop, D. R., Kangasluoma, J., and Ehn, M.: A Combined Gas- and Particle-phase Analysis of Highly Oxygenated Organic Molecules (HOM) from α-pinene Ozonolysis, EGUsphere, 2022, 1-30, 10.5194/egusphere-2022-1317, 2022.